# A Room-Temperature Surface Acoustic Wave Ammonia Sensor Based on rGO/DPP2T-TT Composite Films

**DOI:** 10.3390/s22145280

**Published:** 2022-07-14

**Authors:** Tien-Tsan Hung, Mei-Hui Chung, Jiun-Yi Wu, Chi-Yen Shen

**Affiliations:** 1Department of Chemical Engineering, I-Shou University, Kaohsiung 84001, Taiwan; tthung@isu.edu.tw; 2Office of Institutional Research, I-Shou University, Kaohsiung 84001, Taiwan; mhchung@isu.edu.tw; 3Department of Electrical Engineering, I-Shou University, Kaohsiung 84001, Taiwan; s7762639@gmail.com

**Keywords:** surface acoustic wave, ammonia, rGO, DPP2T-TT, composite film

## Abstract

Surface acoustic wave (SAW) sensors based on reduced graphene oxide/poly (diketopyrrolopyrrolethiophene-thieno [3,2-*b*]thiophene-thiophene) (rGO/DPP2T-TT) composite sensing films for the detection of ammonia were investigated at room temperature in this study. The rGO/DPP2T-TT composite films were deposited onto ST-X quartz SAW resonators by a drop-casting method. FESEM, EDS, and XRD characterizations showed that the rGO/DPP2T-TT composite film was successfully synthesized and exhibited numerous wrinkles and a rough structure, which are crucial for gas adsorption. The frequency response to 500–1400 ppb ammonia shown by the prepared SAW sensor coated with rGO/DPP2T-TT composite film increased linearly as the ammonia concentration increased. The sensor based on a rGO/DPP2T-TT composite film exhibited a positive frequency shift of 55 Hz/ppm, and its frequency response to 500 ppb ammonia was 35 Hz. The sensors thus show promising potential in detecting sub-ppm concentration levels of NH_3_ at room temperature, which opens up possibilities for applications in the noninvasive detection of NH_3_ in the breath. As a result, the rGO/DPP2T-TT composite sensor can be a good candidate for in situ medical diagnosis and indoor/outdoor environment monitoring.

## 1. Introduction

High technological and industrial development leads to an improved standards of living and provides convenience for daily life. However, it also exacerbates environmental pollution, including that of water, air, and soil. Air pollution affects people’s health the most directly; harmful substances in the air, such as suspended matter, ammonia, sulfide, and nitrogen oxide, enter the body through the lungs, causing major and direct damage to the heart and blood vessels [1,2]. Therefore, the development of gas sensors that are reliable and immediately responsive to harmful substances is urgently required.

One hazardous gas is ammonia, a colorless and poisonous gas, which is generated by ammonification and combustion [3]. Ammonification is mainly performed by microbial metabolic activities, which decompose organic nitrogen from animals and plants. Combustion from the chemical industry, which produces fertilizers and other chemicals, generates ammonia in the atmosphere. Exposure to high concentrations of gaseous ammonia can severely irritate human respiratory organs, skin, and eyes. Meanwhile, ammonia is also generated by human metabolic activity. Ammonia in exhaled breath is a diagnostic biomarker for disturbed urea balance caused by kidney disorders or ulcers caused by Helicobacter pylori bacteria-induced stomach infections [4]. Studies have reported that the concentration of ammonia exhaled by humans is closely associated with helicobacter pylori and end-stage renal disease [4,5]. For example, patients diagnosed with end-stage renal disease were found to exhale ammonia in higher concentrations (4.88 ppm on average) compared with healthy individuals (0.96 ppm on average), and the concentration of ammonia exhaled by patients with helicobacter pylori before medication (0.4 ppm) was eight times as high as that after medication (0.05 ppm) [6]. Patients diagnosed with kidney disease have an inhibited kidney function, preventing normal urea metabolism; consequently, blood urea nitrogen (BUN) increases. In a healthy body, blood contains 7–20 mg of urea per 100 mL; therefore, a BUN is higher than 20 mg/dl suggests problems with kidney function. BUN affects the concentration of saliva urea nitrogen in the human body. Saliva urea nitrogen is hydrolyzed in the oral cavity to generate expiratory ammonia, which can be detected using a highly sensitive ammonia sensor [7,8]. One study has reported ammonia concentrations of 820–14,700 ppb in patients with kidney disease [6]. Accordingly, developing a high-sensitivity, high-selectivity trace ammonia sensor capable of real-time detection of sub-ppm levels is critical for medical diagnosis and environmental monitoring.

Materials commonly employed for ammonia sensing include semiconductor metal oxides, conducting polymers, graphene, and composite materials, which are implemented in various sensor components, such as chemical resistance sensors, surface acoustic wave (SAW) filters, quartz crystal microbalances, optical sensors, and metal oxide semiconductor field-effect transistors. Semiconductor metal oxides have been prominently studied as materials for gas sensors because of their advantages, such as low costs, high sensitivity, rapid response and recovery, satisfactory long-term stability, ease of use, low maintenance costs, and capability to detect most hazardous gases; accordingly, these oxides have been applied in a broad range of fields and have attracted considerable attention. Conducting polymers have been incorporated as gas sensor materials since the 1980s, when polyaniline, polypyrrole, and polythiophene were applied as the sensing layers of gas sensors [9]. In recent years, an increasing number of studies have focused on conducting polymers because they can be studied at room temperature, are highly sensitive, respond to gases rapidly, and are easy to synthesize [10,11,12,13,14]; moreover, their molecular structures can be changed through copolymerization or structural derivation [15]. However, these polymers are poor in terms of long-term stability, reversibility, thermal stability, and selectivity [16]. Poly (diketopyrrolopyrrolethiophene-thieno [3,2-*b*]thiophene-thiophene) (DPP2T-TT), a donor acceptor-conjugated conducting polymer [17,18,19], is one of the most promising polymers for applications in organic field-effect transistors (OFETs) due to the outstanding aggregation properties provided by its strong donor–acceptor interactions and its solution processability. In our previous study, we also proved that SAW sensors coated with DPP2T-TT film can be operated at room temperature and can detect the concentration of ammonia at the parts-per-billion (ppb) level [19]. Reduced graphene oxide (rGO) has increasingly been applied in recent years as the sensing layers of gas sensors because it has numerous surface defects, which enhance its gas adsorption. Moreover, the chemical and electrical properties of rGO change easily; this enables rGO to effectively transfer charges, which enhances its sensing performance. Studies have shown that rGO sensing layers have particularly high sensitivity, selectivity, and stability and are rapid in responses and recovery at room temperature [20,21].

In our previous work [19], we showed that a SAW sensor coated with a DPP2T-TT sensing layer is a potential candidate for applications in the detection of gastric because the ammonia concentration in exhaled breath is between approximately 50 to 400 ppb in people with helicobacter pylori infection. However, in patients with kidney diseases, the approximate ammonia concentration in exhaled breath is between 0.82 and 14.7 ppm. We therefore must develop a sensor with a frequency response to sub-ppm ammonia that increases linearly as the ammonia concentration increases for applications in the detection of kidney diseases. DPP2T-TT sensing film has a smaller specific surface area, leading to a limited amount of adsorbed gas, and a relationship between frequency shift and concentration cannot be a linear in higher concentration range, so the formulation of the sensing films must be modified. In this study, a composite film created using DPP2T-TT with a high sensitivity blended with rGO with a large specific surface area was employed to detect ammonia in the sub-ppm range. The film was jointly applied with a low-loss SAW resonator created using an electrode-width-controlled/single-phase unidirectional transducer (EWC/SPUDT) to create a high-sensitivity sensor capable of detecting sub-ppm ammonia for being a good candidate for kidney diseases detection applications and indoor/outdoor environmental monitoring. The linear response, sensitivity, repeatability, reversibility, and selectivity of the sensor were examined to verify that its sensing properties were satisfactory, thereby simplifying the complex and time-consuming in situ detection of sub-ppm ammonia from non-invasive exhaled breath analysis in the future.

## 2. Experimental Methods

### 2.1. Device Fabrication

The SAW device used in this study was based on an ST-X quartz substrate and was fabricated using semiconductor lift-off and sputtering technology. A dual-track structure was adopted to reduce the effects of environmental factors. The input/output interdigital transducer (IDT) in each channel adopted the EWC/SPUDT structure and was combined with the reflection grating on both sides of the channel to form a two-port resonator [19]. Figure 1 presents a magnified image of the EWC/SPUDT structure, in which the width of the wide electrode was λ/4; the width of the narrow electrode was λ/8; the distance between the wide and narrow electrodes was 3λ/16; and the distance between the two narrow electrodes was λ/8. The center of activation was positioned 3λ/8 to the left of the wide electrode center, enhancing the power of the positive signal and effectively reducing the insertion loss [22,23]. The operating frequency of the SAW sensor was 98.5 MHz with an IDT period of 32 μm; the number of pairs of input and output EWC IDTs was 106, and the overlapping length was 1 mm (31λ); the delay-line distance between each pair of input and output IDTs was 600 μm; there were 150 strip gratings on each of the two sides [19]. The IDTs and gratings were aluminum, and the size of the dual-track structure was 15.3 × 5.8 mm^2^. Figure 2 presents a top view of the EWC/SPUDT SAW sensing chip employed in this study.

### 2.2. Preparation of rGO/DPP2T-TT

The GO was prepared using the Hummers method: 5 g of graphite powder (Acros, 99%), 2.5 g of K_2_S_2_O_8_ (Alfa Aesar, 99%), 2.5 g of P_2_O_5_ (Showa, 98%), and 20 mL of H_2_SO_4_ (Echo, 98%) were mixed in a 50 mL Erlenmeyer flask, stirred at 80 °C for 6 h, and cooled and stirred for an entire night. The preoxidized graphite suspension was mixed with 5 g of NaNO_3_ (Alfa Aesar, 98%) and 50 mL of H_2_SO_4_ in a 500-mL volumetric flask, which was then placed in an ice bath. The mixture was stirred at 0–5 °C for 30 min and had 30 g of potassium permanganate (Showa, 99%) slowly added without the temperature being raised beyond 15 °C. The mixture was then removed from the ice bath and stirred at 35 °C for 2 h. The suspension was diluted with 230 mL of deionized water and then immediately stirred at 98 °C for 2 h. The solution was then diluted with 100 mL of deionized water and stirred consistently; 50 mL of hydrogen peroxide (Showa, 30 wt%) was added to remove the excess potassium permanganate. The mixture was rinsed using 1% HCl (J.T.Baker, 37%), centrifuged, and rinsed to the natural pH value using deionized water. The solution was then filtered and dried in a vacuum at 55 °C to extract the GO.

Subsequently, 300 mg of GO was extracted and dispersed in 100 mL of deionized water through an ultrasonic bath. The solution had hydrazine hydrate (Fisher UK, 99%) added at 1 mL per 100 mg of GO and left to react at 95 °C for 1 h to create rGO. The rGO was collected through filtering and rinsed using deionized water to remove the excess hydrazine hydrate. The rGO was then dried naturally and sieve-analyzed in a 60 °C vacuum oven to acquire the black rGO powder.

Then, 50 mg of DPP2T-TT (Sigma-Aldrich, St. Louis, MI, USA) was slowly added to 5 mL of chlorobenzene (C_6_H_5_Cl; TEDIA, Fairfield, OH, USA). The solution had to be stirred slowly to enable DPP2T-TT to fully dissolve in the chlorobenzene. Subsequently, the solution underwent a 3 min ultrasonic bath for thorough dissolution, forming a dark-colored DPP2T-TT solution. Then, 5 mg of rGO was added to the solution while it was stirred, creating an rGO/DPP2T-TT solution. A sensing track was produced by drop-coating a 1.5 × 0.5 mm^2^ area of the layer of rGO/DPP2T-TT in between the input/output IDTs, and the reference track surface was kept free. The dual-track SAW sensing chip was annealed in an oven at 80 °C for 1 h to obtain a stable rGO/DPP2T-TT composite film. Finally, the sensing chip was annealed with the oscillator circuit to develop the SAW sensor.

### 2.3. Experimental Setup

The target gas for the sensor was NH_3_, with high-purity dry air (Jing De Gases, Kaohsiung, Taiwan) as the carrier gas. The temperature of the experiment was 24 °C. A mass flow controller (5850E, Brooks, PA, USA) was applied to control the concentration of the analyte. The total flow of all gases combined was controlled at 110 mL/min. The gas control valve was used to control the input and output of the analyte. Finally, a frequency counter (53132A, Agilent, CA, USA) was applied to measure the frequency changes and was connected to a computer to acquire the experimental data (Figure 3). At the beginning of the experiment, the sensing chip was kept in a vacuum chamber at 65 °C for 1 h to remove all absorbate and unknown small molecules from the sensitive film. The chip was then placed in the sealed 5 cm^3^ test chamber. Dry air was first introduced into the test chamber for 30 min, as demonstrated by the red arrow (Arrow 1) in Figure 3, to eliminate unwanted gas molecules and to stabilize the background signal. With dry air as the carrier gas, the analyte (NH_3_) had its concentration controlled for sensing using the mass flow controller, and the gases were mixed for a minimum of 30 min to ensure their even mixture. The gas valve was then rotated to introduce the analyte into the test chamber for 3 min of sensing, as depicted by the green arrow (Arrow 2) in Figure 3, followed by a dry air purge for 30 min. After the experiment, the sensing chip was preserved in a sealed box filled with nitrogen to prevent contamination with impurities or moisture in the air. The sensor response is defined as the variation in the operating frequency of the SAW sensors due to NH_3_ adsorption. The frequency responses in all experiments were recorded using the frequency counter and transmitted to the computer through the GPIB interface.

## 3. Results and Discussion

### 3.1. Analysis of the rGO/DPP2T-TT Composite Film

After the rGO/DPP2T-TT solution had been prepared, a pipette was used to extract the solution 0.5 mL at a time and cast it to air-dry on a 0.5 × 0.5 cm^2^ cover glass. The process was repeated five times at 5 min intervals to complete an rGO/DPP2T-TT composite film. A field emission scanning electron microscope (FESEM; Hitachi-4700, Tokyo, Japan) equipped with an energy-dispersive X-ray spectroscopy (EDS; HORIBA, Kyoto, Japan), and an X-ray diffractometer (XRD; PANalytical-X’Pert PRO MPD, Malvern, UK; Cu kα/45 kV/40 mA, λ = 1.5406 Å), were employed to analyze the chemical composition and materials of the film.

Figure 4 depicts 250×, 5000×, and 30,000× FESEM images of the film. As shown in Figure 4a,b, there were numerous holes and wrinkles on the film’s surface, and layered accumulation is the typical form of rGO. This indicated that rGO and DPP2T-TT had been satisfactorily synthesized, and the film exhibited a satisfactory rough surface. Wrinkles were available as adsorption sites in rGO and would induce adsorption, which could enable effective gas sensing through an increased surface area for gas adsorption [24,25]. The 30,000× image in Figure 4c is a magnified image of the area highlighted with a red frame in Figure 4b and depicts various wrinkle patterns on the film surface, which also increased the overall adsorption surface area, enabling effective ammonia adsorption.

Figure 5 depicts the EDS mapping of the rGO/DPP2T-TT composite film, which displays the element’s distribution because DPP2T-TT is formed through the polymerization of diketopyrrolopyrrole (C_6_H_2_N_2_O_2_), thiophene (C_4_H_5_S), and pyrrole (C_4_H_5_N). Therefore, in the EDS image (Figure 5), carbon was provided by these three chemical substances and rGO, oxygen was provided by diketopyrrolopyrrole, nitrogen was contributed by pyrrole, and sulfur was provided by thiophene.

Figure 6 illustrates the XRD analysis on the rGO/DPP2T-TT composite film, which reveals a visible and intense diffraction peak between 3° and 5°. According to a previous study [19], the peak was caused by the chemical structure side chain of DPP2T-TT. A broadened diffraction peak was also noted between 15° and 30°; this peak was caused by rGO [26]. Thus, an rGO/DPP2T-TT composite film was successfully prepared.

### 3.2. Mechanism of Gas Sensing

SAW devices were coated with a sensing layer for chemical sensing. Any changes in the mass, mechanical, or electrical properties of this sensing layer upon exposure to the foreign molecules can perturb the surface acoustic waves enabling the devices to be used as sensors [27]. The perturbation from the wave propagation characteristics after gas adsorption can be written as:(1)$$\frac{\Delta f}{f_{0}}≅\frac{\Delta v}{v_{0}}=-c_{m}f_{0}\Delta \left(\frac{m}{A}\right)+4c_{e}f_{0}\left(\Delta hG^{’}\right)-\frac{K^{2}}{2}\Delta \frac{1}{{\left(v_{0}C_{s}/\sigma_{s}\right)}^{2}+1}$$
where $c_{m}$ and $c_{e}$ are the coefficients of mass sensitivity and elasticity, respectively; m/A is the change in mass per unit area; h is the thickness of the sensing layer; G’ is the shear modulus; $K^{2}$ is the electromechanical coupling coefficient; $\sigma_{s}$ is the sheet conductivity of the sensing layer; *C_s_* is the capacitance per unit length of the SAW substrate; *v*_0_ and *f*_0_ are the wave velocity and the operating frequency of the SAW sensor before target gas sensing, respectively; and $\Delta f=f-f_{0}$. The surface acoustic wave interacts mechanically with mass and elasticity, and, consequently, its frequency changes with them. According to Equation (1), the frequency shifts negatively and positively with the mass and elasticity loadings, respectively. The third term on the right hand of Equation (1) represents the acoustoelectric effect. The frequency shift can reflect the sensitivity of a SAW gas sensor.

The gas-sensing mechanism of the rGO/DPP2T-TT composite film to NH_3_ gas in dry air at room temperature is not fully understood at present. Several sensing mechanisms have been proposed for the rGO/conducting polymer composite systems, including redox reactions between the composite film and the gas analyte, the charge transfer between the composite film and the gas analyte, the composite film swelling and the rGO/conducting polymer may form a p–n junction [28]. In our system, we presume that the direct charge transfer process between NH_3_ molecules and rGO/DPP2T-TT composite film surface appears to be the most probable dominant process. When NH_3_ molecules are adsorbed on the rGO/DPP2T-TT composite film surface by physisorption, the holes of the conductive rGO/DPP2T-TT composite film will interact with the electron-donating NH_3_ analyte. The delocalization degree of the conjugated π electrons of the sensing film is increased by the charge transfer from the adsorbed NH_3_ molecules. This leads to the formation of a neutral polymer backbone and a decrease in the charge carriers resulting in a decrease in the electrical conductivity of the sensing film. As a result, a positive frequency shift is observed. Moreover, the addition of rGO into DPP2T-TT increases the specific adsorption surface area and π–π interactions leading to improved sensing performance of the SAW sensor.

### 3.3. Gas Sensing Properties

The gas sensor properties investigated in this study were linearity, sensitivity, reversibility, repeatability, long-term stability, response time, recovery time, and selectivity. Figure 7 shows the frequency responses of the proposed SAW sensor to different concentrations of ammonia in dry air. Each data point was calculated as the mean of the results of three experiments conducted at the same ammonia concentration. The bars in Figure 7 represent the standard error. The frequency response stably and linearly increased following a rise in the ammonia concentration. Therefore, this SAW sensor can be used to effectively detect 500–1400 ppb of ammonia in dry air at room temperature. The sensing mechanism indicates that the rGO/DPP2T-TT composite film exhibited an increase in resistance when the film was exposed to NH_3_ gas. This made caused the third term on the right-hand of Equation (1) to result in a positive frequency [29]. According to the results indicated in Figure 7 and the perturbation mechanism demonstrated in Equation (1), the positive frequency shifts indicate that the sum of the elastic and acoustoelectric effects when the rGO/DPP2T-TT composite film adsorbed ammonia was higher than the mass loading.

Table 1 lists the frequency responses, response time, and recovery time of the proposed SAW sensor to different concentrations of ammonia in dry air. The response time refers to the time required to raise the frequency response to 90% of its maximum after the analyte input. The recovery time refers to the time required to return the frequency response to 90% of its baseline after the analyte removal. The sensitivity is expressed as follows:(2)$$\mathrm{Sensitivity}=\frac{\Delta f}{\Delta C}$$
where Δ*f* indicates the change in frequency response, and Δ*C* indicates the change in ammonia concentration. According to the data listed in Table 1, the sensitivity of the SAW sensor was 55 Hz/ppm. A slightly longer period was required for this SAW sensor to return to its initial value after the NH_3_ gas concentration decreased to zero; according to [30], this is because the strong bond between the rGO and ammonia increased the time required. The minimal detectable concentration for the sensor was the concentration detected at a signal-to-noise ratio of 3. When the sensor detected 500 ppb ammonia, its frequency response was 35 Hz and the noise level was 9.9 Hz, so the signal-to-noise ratio was calculated as 3.53. Accordingly, the estimated detection limit of the proposed SAW sensor was 425 ppb NH_3_ at a signal-to-noise ratio of three.

In addition to sensitivity, determining whether a gas sensor is capable of satisfactory performance and development requires the examination of its repeatability. Ideally, when a sensor is repeatable, it generates the same frequency response when the same operation method and analyte gas are repeated. In this study, 750 ppb ammonia was implemented in the six-cycle consecutive test for the proposed SAW sensor. As shown in Figure 8, the frequency response was 59 Hz in the first cycle and 55 Hz in the sixth. The repeatability is expressed as follows:(3)$$\mathrm{Repeatability}\;(\%)=\frac{f_{6}}{f_{1}}\times 100\%$$
where *f*_1_ represents the frequency response in the first cycle and *f*_6_ represents that in the sixth cycle. The proposed SAW sensor exhibited 93% repeatability in the sixth cycle, indicating that it is satisfactorily repeatable when exposed to 750 ppb of ammonia.

Long-term stability is also related to the reliability of a gas sensor; it reveals whether the material changes over time and affects the experiment results. In this study, the proposed SAW sensor was exposed to 1400 ppb ammonia for 31 days, and its measurement results were recorded every 10 days (Figure 9). The frequency response dropped from 110 Hz on the first day to 60 Hz on the 31st day, indicating the long-term stability of the rGO/DPP2T-TT composite film to be unsatisfactory. This is a problem that must be addressed in further gas sensor development. Long-term stability is expressed as follows:(4)$$\mathrm{Long-term}\;\mathrm{stability}\;(\%)=\frac{f_{D31}}{f_{D1}}\times 100\%$$
where $f_{D1}$ represents the frequency response on the 1st day, and $f_{D31}$ represents that on the 31st day. In this study, the frequency response measured on the 31st day was only 54% of that on the 1st day.

Selectivity refers to whether a gas sensor can respond to one specific target gas without being influenced by other gases. In this study, 1 ppm CO_2_, 1 ppm CO, and 1 ppm H_2_ were employed as interfering gases. Figure 10 illustrates the results of the selectivity test, with the vertical axis (Hz/ppb) indicating the sensitivity of the sensor to various gases. It is clearly seen from Figure 10 that the sensitivity of the proposed SAW sensor to 600 ppb NH_3_ gas is over two times higher than 1 ppm CO_2_ and 1 ppm H_2_, but the sensitivity to 1 ppm CO shows negative frequency shifts. Accordingly, the SAW sensor coated with rGO/DPP2T-TT composite film is capable of detecting ammonia at sub-ppm concentration levels in dry air at room temperature when interfering gases are present.

## 4. Conclusions

This study employed a SAW sensor coated with rGO/DPP2T-TT composite film to detect ammonia at the sub-ppm concentration levels in dry air to determine its potential for applications in biomedical sensing. As shown in the FESEM, EDS, and XRD images, the rGO/DPP2T-TT composite film was successfully synthesized and exhibited numerous wrinkles and a rough structure, which is crucial to gas detection. According to the experimental results, the frequency response of the prepared SAW sensor coated with rGO/DPP2T-TT composite film to 500–1400 ppb ammonia increased linearly as the ammonia concentration increased. The sensitivity of the SAW sensor coated with the rGO/DPP2T-TT composite film was calculated as 55 Hz/ppm, and its frequency response to 500 ppb ammonia was 35 Hz, indicating that its minimal detectable ammonia concentration was 425 ppb. According to its recovery time, this sensor did not exhibit excellent reversibility because the strong bond between the ammonia and rGO rendered desorption difficult. The repeatability of the SAW sensor coated with the rGO/DPP2T-TT composite film was 93%. However, its frequency response on the 31st day was only 54% of that on the first day, indicating that its long-term stability poses a major obstacle to its further development, which warrants further research. The SAW sensor was highly selective to H_2_, CO, and CO_2_ at the sub-ppm concentration levels. In summary, this study’s SAW sensor coated with rGO/DPP2T-TT composite film exhibited satisfactory linear responses, selectivity, and repeatability to ammonia at sub-ppm concentration levels, but its sensitivity, reversibility, and long-term stability require further improvement. Future studies will also explore the sensing properties of SAW sensors coated with rGO/DPP2T-TT composite sensing layer under various humidities for the practical application of breath tests.

## Figures and Tables

**Figure 1 sensors-22-05280-f001:**
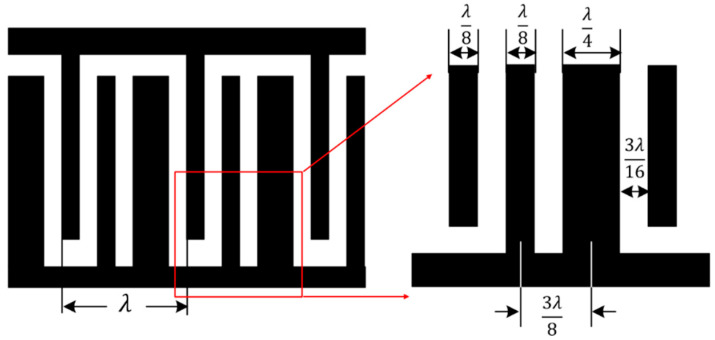
A magnified image of the EWC/SPUDT structure.

**Figure 2 sensors-22-05280-f002:**
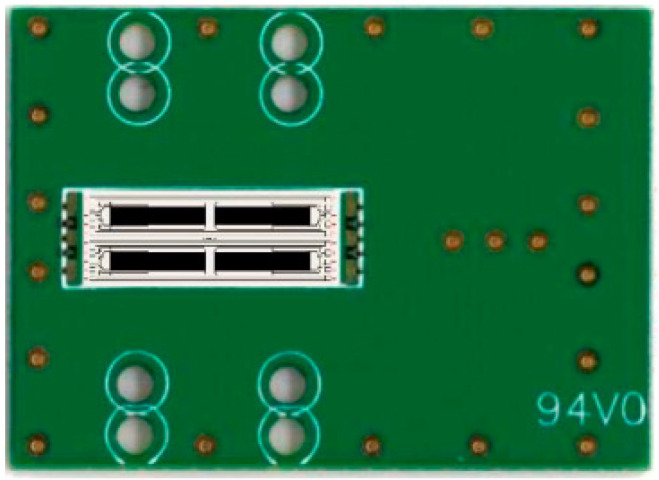
The EWC/SPUDT SAW sensing chip.

**Figure 3 sensors-22-05280-f003:**
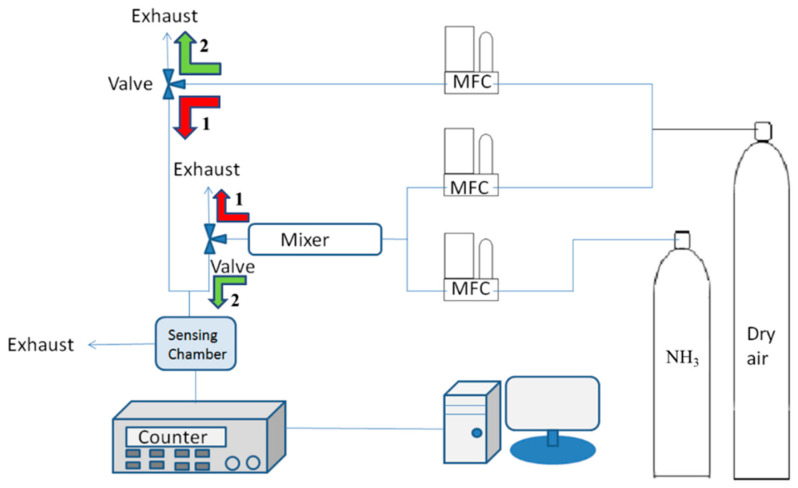
Experimental setup for ammonia gas sensing measurement.

**Figure 4 sensors-22-05280-f004:**
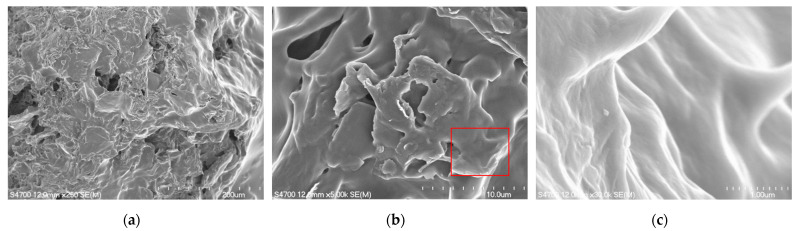
Top view FE-SEM images of the rGO/DPP2T-TT film: (**a**) 250×; (**b**) 5000×; and (**c**) 30,000×.

**Figure 5 sensors-22-05280-f005:**
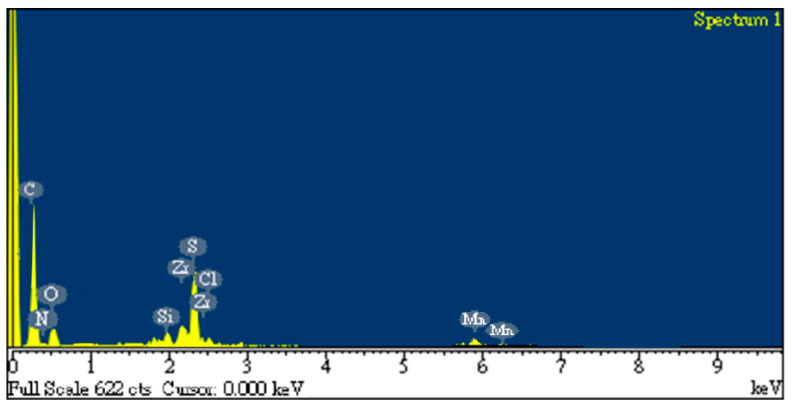
EDS mapping of the rGO/DPP2T-TT composite film.

**Figure 6 sensors-22-05280-f006:**
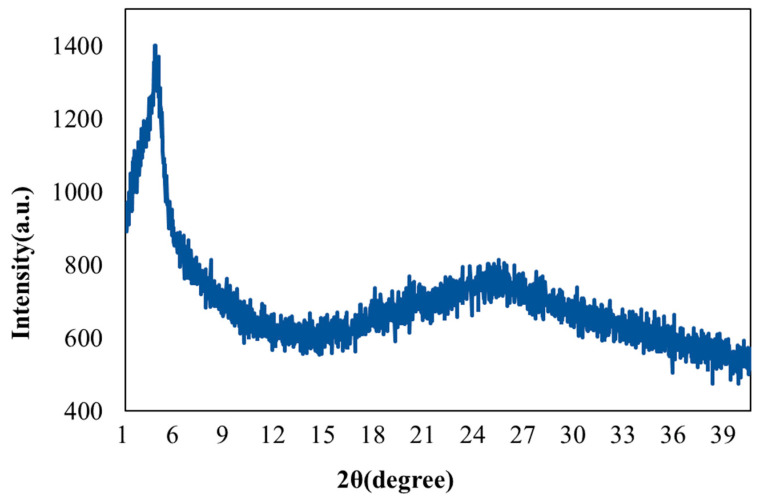
X-ray diffraction pattern of the rGO/DPP2T-TT film.

**Figure 7 sensors-22-05280-f007:**
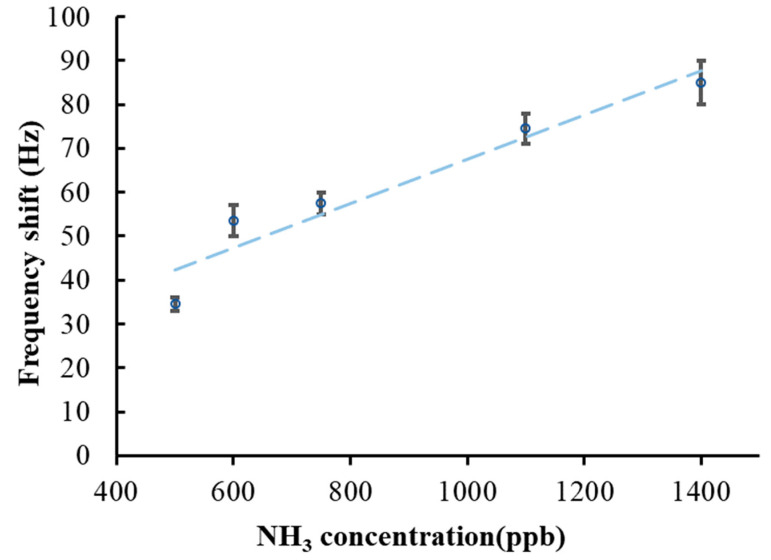
The frequency shift of a SAW sensor as a function of ammonia concentrations in dry air; the R^2^ coefficient of determination is 0.9238.

**Figure 8 sensors-22-05280-f008:**
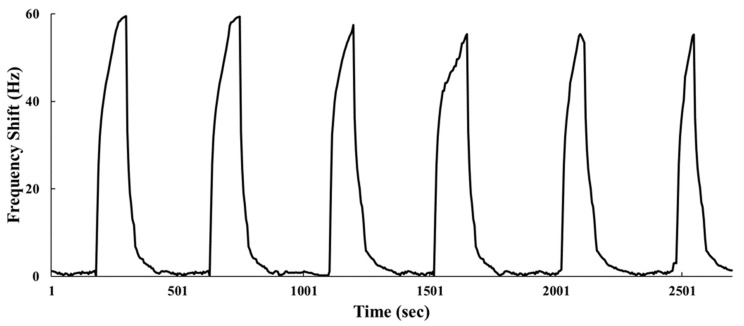
Dynamic responses of a SAW sensor to 750 ppb NH_3_ for six consecutive cycles.

**Figure 9 sensors-22-05280-f009:**
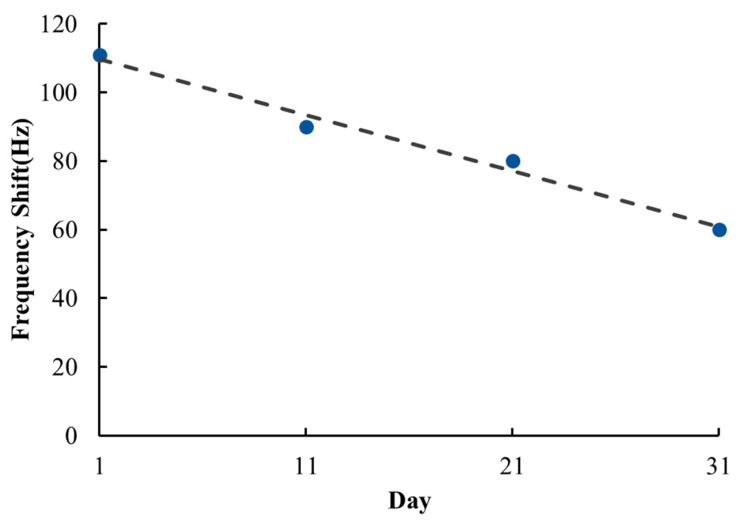
Responses of a SAW sensor to 1400 ppb NH_3_ gas within 31 days.

**Figure 10 sensors-22-05280-f010:**
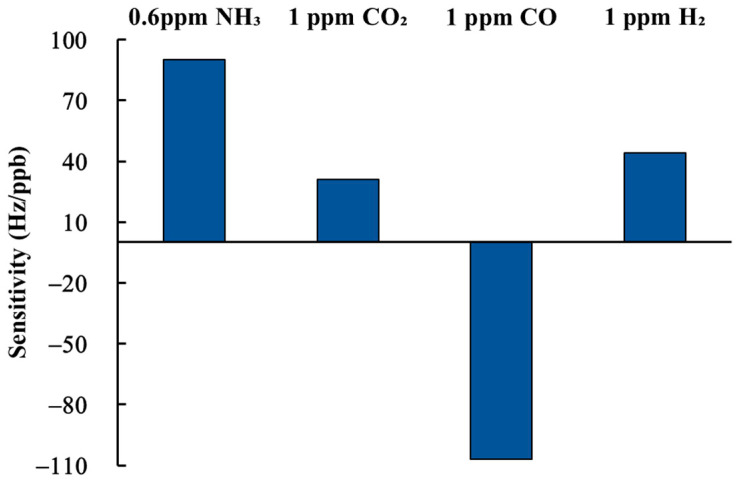
Sensitivity of a SAW sensor to 0.6 ppm NH_3_ gas, 1 ppm CO_2_ gas, 1 ppm CO gas, and 1 ppm H_2_ gas.

**Table 1 sensors-22-05280-t001:** Sensing performance of the SAW sensor toward NH_3_ gas in dry air.

NH_3_ Concentration (ppb)	500	600	750	1100	1400
Frequency shift (Hz)	35	54	58	75	85
Response time (s)	30	57	35	130	112
Recovery time (s)	60	68	60	248	136

## Data Availability

Not applicable.

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
