# Peer review of "A Room-Temperature Surface Acoustic Wave Ammonia Sensor Based on rGO/DPP2T-TT Composite Films"

_sensors, 2022, doi:10.3390/s22145280_

Round 1

Reviewer 1 Report

This manuscript presents studies of surface acoustic wave sensors based on reduced graphene oxide and thiophene compounds sensitive to ammonia at room temperature.

The research methodology was chosen, basically, correctly. However, the goals stated in the introduction do not coincide with the results presented in the manuscript. A number of statements are not proven. In this regard, there are significant comments to the manuscript.

1. It is known that the range of the measured concentration of gas sensors, as a rule, is equal to one order of magnitude in concentration. For example, from 500ppb to 5000ppb. The manuscript investigated the range of measured concentrations of ammonia only from 500ppb to 1400ppb. This range does not meet the requirements for an industrial gas sensor. The authors need to conduct additional studies with concentrations of ammonia up to 5000ppb.

The authors previously, in the work 10.1016/j.sse.2021.108191, developed a more sensitive ammonia sensor based on thiophene. But they do not provide a comparative analysis with the sensor presented in this manuscript.

L27-39 These proposals are not relevant to the material of the article.

L132-144 It is necessary to specify the characteristics of chemical reagents (manufacturer, degree of purity).

L162-183. According to the data presented in section 2.3, the gas flow rate should be about 0.5 – 1 cm/s. This is a very big speed. At this speed, the sensor may cool down. How was the temperature of the studies taken into account? In addition, the authors write that the sensor in the test chamber was kept in nitrogen, and then dry nitrogen was fed into the chamber. How the oscillation frequency changed when the sensor was in nitrogen and in dry air? This is important to know in order to determine the mechanism of reactions on the sensor surface. The authors write about this in section 3.2. Main note: the authors in the introduction describe in detail the concentrations of ammonia in the exhalation of sick people. However, there is a large amount of moisture in the exhalation of people. Why don't the authors investigate the sensor for the effects of moisture?

L198-199 The expression is not proven. The authors do not investigate porosity and active surface area. On the basis of which measurements the authors write that the porosity is satisfactory and the surface area is increased?

L202 The expression has not been proven. The authors do not investigate the adsorption of ammonia.

L209-213. In my opinion, it is not enough to claim that an rGO/DPP2T-TT composite film has been obtained only on the basis of one X-ray diffraction pattern. The authors give a fuzzy reference to the corners of 2 Θ.  In addition, in the work (http://dx.doi.org/10.1016/j.elspec.2014.07.003 ) shows an angle of 2Θ=42.74â—¦ for the rGO film. It is necessary to conduct additional studies using Raman spectroscopy or XPS studies.

L214 in Fig.4 (a,b,c), it is necessary to specify the measuring ruler and numbers more clearly.

L224 Not all variables are specified for equation (1).

 L233-238 The statements made require confirmation in other literary sources or additional measurements. See the remark above.

L243-244 Link [26] is unsuccessful. The article [26] provides references to the following articles to describe the reaction mechanism of graphene with ammonia ([37] Khun Khun K, Mahajan A, Bedi RK. SnO 2 thick films for room temperature gas sensing applications. J Appl Phys 2009;106:124509.

[38] Ghimbeu C, Schoonman J, Lumbreras M, Siadat M. Electrostatic spray deposited zinc oxide films for gas sensor applications. Appl Surf Sci 2007;253:7483–9.

[39] Lam SK, Chan MA, Lo D. Characterization of phosphorescence oxygen sensor based on erythrosin B in sol–gel silica in wide pressure and temperature ranges. Sensor Actuat B: Chem 2001;73:135-41.).

But these articles talk about metal oxides, not graphene. On metal oxides, really adsorbed oxygen molecules become ions. But on the surface of graphene, everything can be different. The authors need to prove the presence of O2 ions- or find new literary sources.

I believe that formula (2) is unproven. 

L254 It is necessary to estimate the measurement error. What is the error of linear approximation in Fig.8? In Fig.8, the error is not defined for all points.

L291-291 is a very unstable repeatability for only two dimensions.

L298-304 is a really unstable sensor.

L319-339 In connection with the comments made, the conclusions require serious adjustments.

Reviewer 2 Report

A SAW ammonia sensor using rGO/DPP2T-TT Composite Films was proposed in this work. Some meaningful results were reported. Some issues should be considered prior to acceptation.

1.     The measured frequency response of the SAW device for sensing ammonia should be added in the manuscript.

2.     The stability (repeatability) was evaluated by only two-cycle consecutive test, that is not enough, over five cycles should be provided.

3.     The selective detection of ammonia is also another important factor to evaluate the sensing performance. The sensing response to other major interferer gases, like H2S should also be considered.

4.     Humidity in testing environment may produce significant in gas sensing. The author should consider the compensation of the crossed humidity sensitivity.

5.     As shown in Table 1, the response time and recovery time of 1100ppb ammonia is longer than that of 1400ppb ammonia. Can the author explain the reason for this phenomenon?

6.     The innovation of this paper should be clarified, and a comparison with previous work should be given.

Round 2

Reviewer 1 Report

The authors were unable to respond to some of my comments

1. Fig.4c shows the morphology of the surface, not the porosity of the material and the adsorption area of its surface.  These parameters are measured using the adsorption of pure gases based on the BET method.

2. In Fig.7, the frequency measurement error was not estimated for the 500 ppb point.

3. L305. Expression (3) should now be expression (2)

4. It is necessary to remove the extra image in Fig.8

Reviewer 2 Report

The author answers well for the initial comments.

Some minor methodological errors and text editing should be further performed. 
